# Training Petting Zoo Sheep to Act Like Petting Zoo Sheep: An Empirical Evaluation of Response-Independent Schedules and Shaping with Negative Reinforcement

**DOI:** 10.3390/ani10071122

**Published:** 2020-07-01

**Authors:** Eduardo J. Fernandez

**Affiliations:** School of Behavior Analysis, Florida Institute of Technology, Melbourne, FL 32901, USA; efernandez@my.fit.edu or edjfern@gmail.com; Tel.: +1-206-765-7350

**Keywords:** animal training, counterconditioning, husbandry, negative reinforcement, operant conditioning, petting zoo, response-independent schedules, sheep, shaping

## Abstract

**Simple Summary:**

The present study examines the use of response-independent food schedules (food delivered independent of what an animal is doing) and negative reinforcement in the form of a trainer removing themselves from the presence of an animal to effectively train approach and contact behaviors in petting zoo sheep. All sheep had previously been reported to avoid/escape the presence of zoo staff and caretakers, as well as refuse to eat in the presence of any persons. In Experiment 1, we were able to effectively use response-independent schedules to produce both approach and contact behaviors in a Hampshire sheep. In Experiment 2, a Jacob sheep was trained for approach/contact via a shaping procedure that initially rewarded approach behaviors by removing the presence of the trainer (negative reinforcement), and was later switched to rewarding responses directly with food (positive reinforcement) once the sheep began to regularly eat in the trainer’s presence. The results are discussed with respect to how both procedures are effective in examining the applied approach/avoidance behaviors with zoo animals, as well as the greater ethical considerations involved in using negative vs. positive reinforcement to train animals.

**Abstract:**

Shaping through differential reinforcement of successive approximations to a target response has been a cornerstone procedure for the training of novel behavior. However, much of how it has traditionally been implemented occurs through informal observation, rather than any direct, systematic measurement. In the present study, we examine the use of response-independent food schedules and shaping for increasing approach and contact behaviors in petting zoo sheep. In Experiment 1, a fixed-time (FT) 15 s food schedule was used to effectively increase approach and contact behaviors in one sheep. In Experiment 2, negative reinforcement in the form of removal of the presence of a trainer was made contingent on the successful completion of approximations within a shaping procedure and later switched to food rewards. A changing-criterion design was used to empirically examine the effects of the shaping procedure during each step of the program. The result is one of the first studies to demonstrate the utility of using negative reinforcement within a shaping procedure to successfully intervene on approach/avoidance behaviors in an applied animal setting.

## 1. Introduction

Shaping has been described as the differential reinforcement of successive approximations to a target response [1,2]. The first use of the word “shaping” occurred in Skinner’s [3] article, “How to Teach Animals,” although descriptions of the procedure were previously given (see [4] for a review). Conceptually, shaping has been considered one of the most important ways of producing behavior change and was among the first applications of behavioral principles outside the laboratory [5].

The use of shaping has become increasingly important for the training of management and husbandry procedures within zoos and similar captive animal facilities [6,7,8]. Some examples of shaping used in zoos and similar facilities include training white-handed gibbons (*Hylobates lar*), Diana monkeys (*Cercopithecus diana*), and a mandrill (*Mandrillus sphinx*) to operate different manipulandum within their enclosures [9]; training an African leopard (*Panthera pardus*) to attend to and pursue acoustic “prey” [10]; moderating aggression, increasing voluntary movement, and obtaining blood and urine samples in chimpanzees (*Pan troglodyte*) [11,12,13]; facilitating veterinary care with reptiles [14]; obtaining blood samples in grizzly bears (*Ursus arctos horribilis)* [15]; training several Bongo (*Tragelaphus eurycerus*) and Nyala (*Tragelaphus angasi*) to reliably enter crates [16,17].

Given the importance of shaping, little has been done to systematically evaluate its implementation, and its practice is often considered more of an “art” than a science [18,19]. According to [19], this is due to “technical difficulties involved in accurately adhering to specific parameters of shaping procedures and in recording the closeness of specific patterns to a given target response” (p. 241). Only a few studies have extended empirical examinations of shaping to include approximations across different response topographies, an often-critical component of many shaping procedures [20,21,22,23,24,25,26].

While shaping often presumes the use of appetitive stimuli such as food to positively reinforce the desired responses, there is a long history of using negative reinforcement to first apply and then remove an aversive stimulus to increase behavior [27,28]. For instance, the field of horse training often uses the removal of pressure from reigns or similar devices to effectively halter train or otherwise shape behaviors [29,30]. Similarly, negative reinforcement has been used to train and maintain keypecking in laboratory pigeons [31]. In addition, a novel approach to shaping using negative reinforcement was proposed by Kurland [32], where the presence of a person was removed contingent on approaching in a llama. In Kurland’s procedure, the trainer would wait for the llama to face and then approach her, deliver a conditioned reinforcer in the form of the click of a clicker, and then move away from the llama. Within several trials, Kurland reported that the llama regularly began to approach her. This differs significantly from counterconditioning procedures such as reciprocal inhibition/systematic desensitization that otherwise rely on gradual exposure of aversive stimuli to reduce fear-based response [33,34,35].

The present study involves training “fearful” sheep (i.e., sheep that avoided people) to engage in approach and contact behaviors with trainers. In Experiment 1, we used response-independent fixed-time (FT) food schedules to train approach and contact responses in a sheep. In Experiment 2, we used negative reinforcement (removal of the trainer’s presence) and positive reinforcement (food delivered) to shape approach and contact responses in a sheep. Experiment 2′s results were examined in terms of (1) the effects across all other steps of the shaping procedure, eating, and undesired responses in the treatment subject, and (2) the effects of the shaping procedure on another sheep that was not directly intervened upon within the shaping procedure.

## 2. General Methods

### 2.1. Subjects, Setting, and Material

Both experiments involved three petting zoo sheep: Oatis, an approximately 5-year-old male Hampshire sheep (Experiment 1), and Fauna and her son Forrest, an approximately 5-year-old female and 1-year-old male Jacob sheep, respectively (Experiment 2), located at the Frank Buck Zoo in Gainesville, Texas. All three sheep were reported to avoid/escape the presence of all zoo staff and caretakers, as well as refuse to eat in the presence of any persons. Sessions were conducted in an 18 × 9 m enclosure. The enclosure normally housed several petting zoo sheep and goats, but for the purposes of the study, all other animals were moved to a different part of the exhibit during study sessions. Session tools included a clicker, a sheep brush, a video camera, several food items that were not part of their daily diet (e.g., bread, wheat thins, and crackers), a food bowl, and a clipboard with data sheets.

### 2.2. Procedure

Two to four persons were involved in any session. The first person (first author) was designated trainer and conducted most of the procedures. A second person was then designated the data collector. Additional persons operated the camera, completed secondary trainer functions, and assisted in data collection. Trials consisted of the start of each trainer–sheep interaction (see below), and sessions consisted of 10 trials. Sessions were conducted one to three times a week, between 09:00 and 15:00, from 27 October 2000 to 8 October 2001. For Experiment 1, all sessions were completed by 20 April 2001, with a follow-up session (6-month probe; see below) collected on 8 October 2001.

#### Response Definitions

On each trial, data were recorded on the completion of each approximation/step met in the procedure, eating, 1–2 trial-terminating (undesired) responses (e.g., “moving away”), and whether no trial-terminating response occurred. Being brushed with a sheep brush was designated the target (final) response, since contacting the brush itself could be required as a prior approximation. In addition, the response of being brushed was compatible with desired petting zoo behaviors of being petted and incompatible with moving away from a trainer or other person. The brushed response was broken down into 7–11 steps (see Table 1):

## 3. Experiment 1

In Experiment 1, we examined the effects of response-independent fixed-time (FT) food schedules on the approach and contact behaviors of a sheep. Fixed-time food schedules involve the delivery of food at “x” amount of time, regardless (independent) of what the animal is doing. We initially implemented this procedure as a control condition for food since food itself could potentially elicit approach responses. However, due to the results observed during the FT schedule, the schedule became the focus of our intervention.

### 3.1. Materials and Methods

The subject of the experiment was Oatis, an ~5-year-old male Hampshire sheep. An ABA reversal design was implemented for the study. Conditions were as follows:

#### 3.1.1. Baseline/Fixed-Time Food Schedule of 15 s 

A response-independent fixed-time food schedule of 15 s (FT-15 s) was used during initial baseline sessions. This involved throwing a piece of food near the sheep regardless of their distance or behavior every 15 s.

Baseline consisted of each step of the procedure being attempted successively until the sheep engaged in moving away, did not meet the criterion for each step within 10 s, or successfully completed all seven steps. For example, the trainer would initiate a trial by announcing, “Start”. If the sheep completed the first step, the trainer would move onto the next step and continue through all steps of the shaping procedure accordingly. If the sheep moved away or did not complete a step within 10 s, the trial would terminate, with a possible new trial shortly following (inter-trial intervals generally lasted several seconds).

During baseline, the clicker was not used, and food was delivered every 15 s (FT-15 s) if a trial was still active. During this and several other conditions (unless otherwise specified), trials began with food being thrown to each sheep (i.e., trial-starting food, recorded as the “eating” response). Termination of a trial resulted in a termination of the FT-15 s schedule, and the time was reset at the start of the next trial. Each session consisted of 10 trials for all conditions in the experiment.

#### 3.1.2. No Fixed-Time Schedule

Because of the success of the FT-15 s schedule in producing all approximations and the target response (see results below), the schedule was removed, and the testing of the approximations and target responses were continued. During this condition, no food was delivered either at the start or during each trial, but the procedure was otherwise run according to the baseline/FT-15 s condition outlined above.

#### 3.1.3. Return to Fixed-Time Schedule (15 s; 30 s; 1 min)

During this condition, the fixed-time schedule was reintroduced. The schedule was thinned from 15 s to 30 s and 1 min accordingly. Starting with the third session of the FT-30 s schedule (Session 18), no food was delivered at the start of the trial, and thus, no “eating” responses were recorded. In addition, during the FT-30 s and FT-1 min schedules, the “eat/hand” step was skipped. 

#### 3.1.4. Fixed-Ratio Schedule (FR-2; FR-5; FR-10)

During this condition, food was no longer delivered on a fixed-time schedule, but instead, “x” number of trials in which the target response was reached were required before food was made contingent on that target response. For instance, in the FR-5 condition, five trials in which each step and the target response were completed was required before a piece of food was delivered. The click of the clicker also followed the successful completion of the schedule and preceded the delivery of the food item. In addition, no food was delivered at the start of the trial, and thus, no “eating” response was possible, and the “eat/hand” step was skipped. 

#### 3.1.5. 6-Month Probe/Fixed-Ratio Schedule (FR-10)

A 6-month probe was implemented to test for retention of the response in the absence of any training. This involved 3 FR-10 sessions, in which all 10 trials were completed in the absence of food before the click of the clicker and food was delivered. As per the previous FR schedules, no food was delivered at the start of the trial, and the “eat/hand” step was skipped.

### 3.2. Results and Discussion

Figure 1 displays the session-by-session data across all conditions for each step in the shaping procedure, eating responses (eating food items that started some of the trials), and the trial-terminating “moving away” responses:

By the third session of the FT-15 s schedule, Oatis ate 100% of the trial-starting food thrown at him and completed 90–100% of the steps and target response of being brushed. Within a few sessions of the FT-15 s schedule, Oatis would bypass some of the initial trial-starting food to be simultaneously brushed and fed every 15 s under this schedule. The removal of the FT-15 s schedule rapidly diminished the completion of most steps in the shaping procedure and increased the moving away to occur more frequently than during the FT-15 s schedule, presumably because no trial-starting food was thrown. The reintroduction of the FT-15 s and later FT-30 s, FT-1 min, and FR schedules successfully maintained completion of the target response during the rest of the experiment, and the 6-month probe demonstrated a 100% retention of the target response.

These results were surprising, given that Oatis, like the other two petting zoo sheep, was reported not to allow staff or other persons to approach him directly. This may have been, in part, due to the fleeing response by the other sheep in the presence of people, and thus partially eliminated once Oatis was separated for the purpose of our study. However, some fleeing responses increased once we eliminated food, and thus, the main factor in our study that contributed to Oatis allowing direct contact by trainers appeared to be the periodic delivery of food.

## 4. Experiment 2

In Experiment 2, we examined the effects of using negative reinforcement in the form of the trainer moving away from the sheep when they engaged in the criterion currently selected. We implemented a fixed-time food schedule of 15 s (FT-15 s) as a control condition to demonstrate both the initially limited consumption of food and approach/contact behaviors targeted for intervention. Only one of two sheep observed in the study, Fauna, was targeted for intervention, with the other sheep, Forrest, used as a control for demonstrating the effects of the shaping procedure. 

### 4.1. Materials and Methods

The subjects of the experiment were Fauna and her son Forrest, an approximately and respectively 5-year-old female and 1-year-old male Jacob sheep. A changing criterion design with multiple reversals was implemented across the different steps of the shaping procedure. All experimental conditions focused on Fauna for intervention, but Forrest was measured simultaneously without directly receiving the intervention. Conditions were as follows: 

#### 4.1.1. Baseline/FT-15 s 

Baseline was the same as Experiment 1. Trial-starting food was thrown to both sheep (i.e., Fauna and Forrest) individually.

#### 4.1.2. Clicker Training-Negative Reinforcement/Escape (CT R−)

During the CT R− condition, a click followed by the trainer moving away from the sheep was delivered for completing the current criterion. For example, if the researchers were currently rewarding “moving within 3 m” (Step D) for intervention, each previous step would be attempted in order and the trial would be continued unless a trial-terminating response occurred. If Step D was successfully completed for that trial, the trainer would deliver a click of the clicker, and then move in the opposite direction of Fauna. During all clicker training conditions, two pieces of food were thrown at the start of the trial and recorded as “eating” for each sheep if they consumed that food item.

#### 4.1.3. Clicker Training–Positive Reinforcement/Food (CT R+)

The CT R+ condition was conducted the same as the CT R− condition, except that a piece of food followed the click of the clicker after the successful completion of that criterion on any given trial. The CT R+ session was implemented in session 36, once Fauna was regularly observed to eat the trial-starting food for several sessions and was maintained throughout the rest of the experiment. The contingency was switched from CT R− to CT R+ because later approximations and the target (final) response involved sheep–trainer contact, which would not have been possible with removal of the trainer delivered as a reward.

### 4.2. Results and Discussion

Figure 2 displays the session-by-session data across all conditions for each step in the shaping procedure, eating responses (eating food items that started some of the trials), and both the trial-terminating responses:

During the initial 10 sessions of the experiment, when the FT-15 s baseline condition was imposed, both Fauna and Forrest exhibited few of the criteria later selected for intervention, few eating responses (10% and 20% of all trials for Fauna and Forrest, respectively), and displayed a high frequency of moving away from the trainer. In addition, the “blocking Forrest” response was only observed during the baseline condition, occurring several times in the first few sessions, but decreasing to 0 for the last three sessions.

On session 11, Step C, “moving toward”, was designated the criterion selected for intervention, and resulted in Fauna completing the step on 90% of all trials. Forrest occasionally followed Fauna, but on two of the three sessions, he spent most trials neither facing nor approaching the trainer. Both sheep also increased their consumption of trial-starting food (e.g., “eating” responses), with Fauna eating approximately 33% of all trial-starting food during sessions 11–13, and Forrest eating 30% of all trial-starting food.

During sessions 14–37, the criterion was cycled through multiple reversals of Step D (standing within 3 m) and Step E (standing within 1.5 m). Fauna did not eat most of the trial-starting food (sessions 14–32; ~31%) and typically would only complete the distance required to approach (3 or 1.5 m) for that trial. Forrest consumed over 50% of the trial-starting food during the same sessions, but only occasionally came within the same distance of the trainer as Fauna. From sessions 33–35, Fauna consumed over 80% of all trial-starting food, therefore beginning with session 36, the contingency was switched from negative reinforcement (CT R−) to positive reinforcement (CT R+). Throughout the rest of the study (sessions 36–99), the positive reinforcement contingency was maintained, and Fauna consumed most of the trial-starting food (~89%), while Forrest only consumed ~7% of all trial-starting food.

From session 38–39, a larger jump in approximations, Step I, “eat/hand”, was attempted unsuccessfully. During both sessions, Fauna generally ended most trials by moving away after completing Step E (standing within 1.5 m).

For the rest of the study (sessions 40–99), approximations were increased one step at a time, with one reversal between Step K, “touch brush”, and Step L, “brushed”, attempted during the latter part of the study. Most trials ended with the criterion selected being successfully completed by Fauna, and the study ended with the target response (Step L; “brushed”) being successfully completed for 29 out of 30 trials (~97%). When trials were unsuccessful, they were typically terminated by the undesired response of moving away from the trainer. In addition, during these latter trials, Forrest rarely engaged in approaching the trainer and spent most of these sessions away from both the trainer and Fauna.

As had been previously observed and confirmed by our baseline condition, both Fauna and Forrest rarely faced/approached trainers or consumed food in a trainer’s presence. The initial negative reinforcement condition was successful at both getting Fauna to approach a trainer and increasing the amount of trial-starting food she would consume. Forrest initially approached trainers and consumed many of the trial-starting food items as well, but eventually decreased in both responses, as the intervention was only directly focused on Fauna. However, this itself was deemed a success, since anecdotal reports from caretakers suggested that Fauna would block Forrest from any human–sheep interactions and leaving her immediate presence.

Overall, the shaping procedure was a success, and the initial inclusion of the negative reinforcement condition allowed for an increase in Fauna’s food consumption. Thus, the ability to use food as a positive reinforcer to shape Fauna’s approach behaviors was facilitated by the prior use of negative reinforcement.

## 5. General Discussion

### 5.1. Response-Independent Schedules and Approach/Contact Behavior

In Experiment 1, we initially imposed a response-independent, fixed-time (FT) food schedule as a baseline control procedure. Similar control procedures have been implemented to account for the effects of reinforcement contingencies [36,37]. However, in our experiment, the procedure resulted in producing the desired approach/contact behaviors in the sheep. It is not clear why this was the case, although other researchers have used “noncontingent reinforcement” (NCR) procedures to decrease aberrant (undesired) responses and increase desired behaviors (for reviews, see [38,39]). Similarly, response-independent schedules have been successful at reducing stereotypic activity and increasing naturalistic foraging responses in zoo animals [40,41].

It is possible that the response-independent schedule functioned as a form of counterconditioning, in which fear-related behaviors were reduced via respondent associations with the food deliveries [42,43]. Similar procedures have been used to produce (rather than reduce) avoidance behaviors in the form of “experimental neuroses” in goats and sheep [44]. Other researchers have proposed a systems-related effect to response-independent schedules, where food presented periodically elicits or affords species-typical foraging behaviors [45,46]. In addition, a similar “open/closed bar” technique has been proposed as a form of counterconditioning to reverse fear and/or aggression in dogs [47]. Regardless, the results were that the procedure was effective at producing the approach/contact behaviors in the Hampshire sheep, and was effectively thinned and maintained 6 months after the intervention. 

### 5.2. Shaping with Negative Reinforcement

In Experiment 2, we were effectively able to use negative reinforcement to shape approach behaviors in a Jacob sheep. At the time of this study, this was one of the first empirical demonstrations of using negative reinforcement in an applied setting to effectively shape approach behaviors, based directly on Kurland’s [32] proposed treatment. Since this time, several other procedures using negative reinforcement with shaping to reduce fearful and/or aggressive behaviors have been established, including constructional aggression treatment (CAT) and behavioral adjustment training (BAT) [48,49,50,51,52,53].

While all the above have procedural differences, they, along with the methods used in this study, share the commonality of using negative reinforcement to effectively shape approach/contact behaviors while simultaneously and effectively reducing fearful and/or aggressive behaviors. This differs substantially from the previously discussed counterconditioning/desensitization procedures used to address fear/aggression, in that (1) they rely on an operant procedure to shape behavior, rather than respondent conditioning associations, and (2) they use the removal of the fear/aggression-inducing stimulus to reward non-fear/aggression-related behavior, rather than some form of periodic exposure to the aversive stimulus.

To date, there are no direct empirical comparisons of shaping-based treatments versus counterconditioning/desensitization procedures to address fear- and/or aggression-related problem behaviors. Therefore, it is not clear what benefits might exist from using operant versus respondent conditioning to address approach/avoidance behavior in applied settings. This is complicated by the fact that desensitization procedures can incorporate operant reinforcement contingencies, thereby combining both types of conditioning [54]. Nonetheless, it would be useful for future studies to directly examine differences between the two procedures and thus provide further evidence for the benefits of using either procedure in treating fear and/or aggression.

### 5.3. Negative Verse Positive Reinforcement and Aversive Control

A point of contention exists in the use of negative reinforcement to influence the behavior of any organism. Namely, that negative reinforcement represents a form of aversive control [55,56]. While others have debated the utility in distinguishing between negative and positive reinforcement [57,58,59], the argument still exists that negative reinforcement is a form of aversive control, and therefore similar to the use of punishment to modify behavior. As a result, guidelines such as the Humane Hierarchy have been developed to address and simplify such ethical considerations [60].

While discussions about the harmful nature of aversive stimuli are ethically relevant and important, it is also worth noting that the use of aversive stimuli is not entirely avoidable nor necessarily desirable to be completely avoided [61,62]. Furthermore, the use of negative reinforcement within this and similar procedures differs substantially from other forms of negative reinforcement used to manage behavior, in that (a) the aversive stimuli presented and then removed were the presence and removal of trainers, and therefore (b) the type of negative reinforcer used was a naturally occurring and generally unavoidable stimulus, as opposed to a contrived aversive stimulus that is continuously presented in a concocted manner to produce the desired behavior change.

The field of behavior analysis has a long-standing tradition of accounting for such uses of aversive contingencies by invoking the Principle of Least Restrictiveness or Least Restrictive Alternative (LRA) [63,64]. In contrast to the Humane Hierarchy, which turns otherwise functionally defined procedures into topographical categories, the Principle of Least Restrictiveness/LRA acknowledges the context of any intervention and makes ethical decisions accordingly. As such, any argument against all aversive control is overly simplistic and does not account for the vast complexity that exists in applied settings. For instance, in our study, it was through using negative reinforcement to shape approach behaviors that we were able to increase food consumption and thus produce the ability to positively reinforce approach and eventual contact behaviors with food. Put simply, to respect the functional, contextual nature of behavior, the argument should not be against all forms of negative reinforcement and/or aversive conditioning, but rather, about the type, intensity, and frequency of the aversive stimuli used and the conditions under which they are applied.

## 6. Conclusions

The present study examined the use of response-independent fixed-time (FT) food schedules (Experiment 1) and a shaping procedure incorporating both negative and positive reinforcement (Experiment 2) on the approach/avoidance behaviors of three petting zoo sheep. In both experiments, the interventions were successful at increasing approach and contact responses and decreasing avoidance responses in the petting zoo sheep, which is a critical behavioral requirement for any petting zoo animal (i.e., the ability to be petted). Both experiments also demonstrated the utility of using quantitative data and within-subject methodology to evaluate the effectiveness of any behavioral intervention.

## Figures and Tables

**Figure 1 animals-10-01122-f001:**
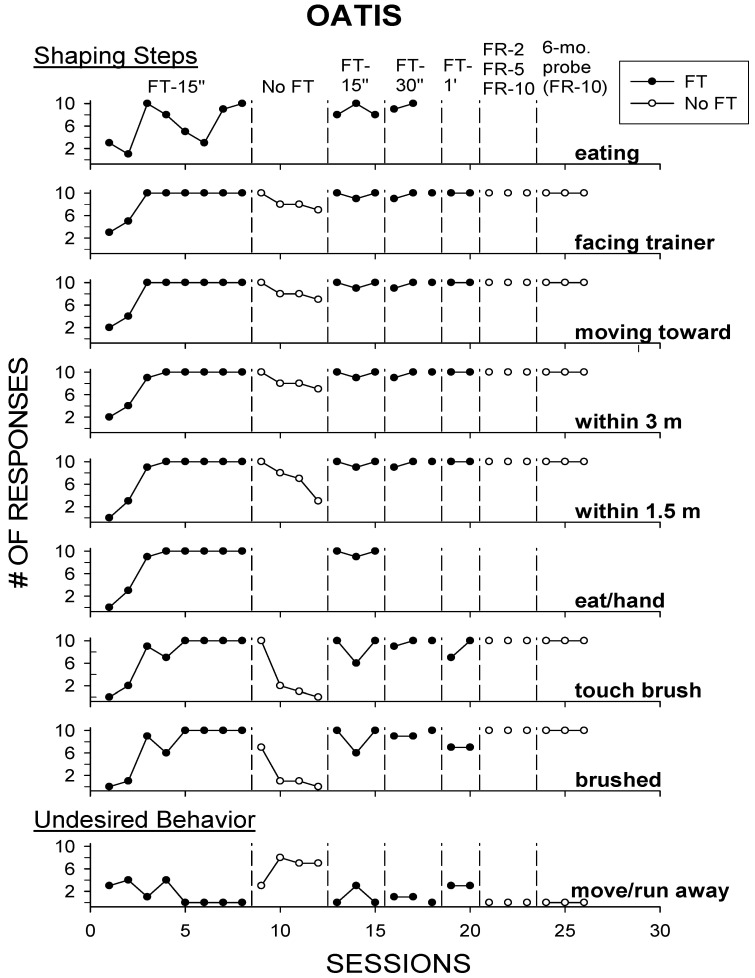
The number of responses for each step in the shaping procedure (top 8 graphs) and the undesired response (moving away; bottom graph) in Experiment 1. Closed circles represent when a Fixed-Time (FT) food schedule was in effect, while open circles represent when no Fixed-Time (No FT) food schedule was in effect. Fixed-Time (FT) schedules of differing times and Fixed-Ratio (FR) schedules of differing responses are listed at the top of the graph. The *y*-axis shows the number of responses that occurred per session (out of 10 trials), and sessions are listed on the *x*-axis.

**Figure 2 animals-10-01122-f002:**
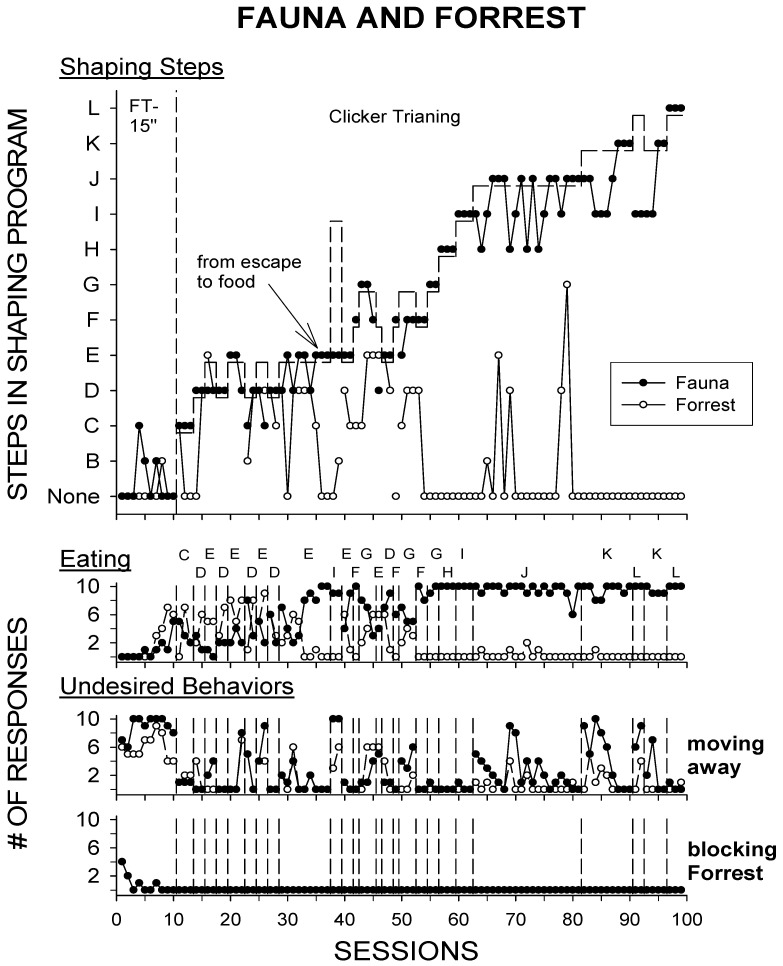
Median final response reached in a trial for each session of the program (top graph), the number of eating responses per session (middle graph), and the number of trial-terminating (undesired) responses per session (bottom two graphs) in Experiment 2. The Baseline condition represents when the Fixed-Time (FT) food schedule of 15 s was in effect. Each step of the Clicker Training shaping procedure is listed on the *y*-axis (see Table 1; B = facing trainer, C = moving toward, D = standing within 3 m, E = standing within 1.5 m, F = standing within 1 m, G = standing within 0.5 m, H = standing within 0.25 m, I = eat/hand, J = contact head/back, K = touch brush, L = brushed), and session number is listed on the *x*-axis. Closed circles represent Fauna’s responses (the treatment animal), while open circles represent Forrest’s responses. The change of the consequence from R− (trainer removal) to R+ (food) occurred with session 36.

**Table 1 animals-10-01122-t001:** Behaviors, definitions, and sessions for the study. The total number of sessions the trial-starting food (e.g., “Eating”) was active (Experiment 1 and 2), as well as the approximation/step intervened upon (Experiment 2) are listed in parentheses.

Behavior/ Step	Definitions	Sessions Exp. 1	Sessions Exp. 2
Eating (thrown food)	Any contact of a sheep’s mouth with food thrown to the sheep at the start of a trial. (Exp. 1 did not always start trial w/food).	1–8; 13–17 (13)	1–99 (99)
Facing Trainer (B)	A sheep’s head faces the direction of the trainer.	N/A	N/A
Moving Toward (C)	The sheep moves in the direction of the trainer.	N/A	11–13 (3)
Standing Within (x) (D-H)	The sheep moves within ‘x’ meters of the trainer. D = 3 m, E = 1.5 m, F = 1 m, G = 0.5 m, H = 0.25 m (Exp. 1 only measured 3 and 1.5 m)	N/A	14–37; 40–59 (44)
Eat/Hand (I)	Any contact of a sheep’s mouth with food being held by the trainer.	N/A	38–39; 60–62 (5)
Contact Head/Back (J)	The trainer contacts the sheep’s head/back without the sheep moving away. (Exp. 2 only)	N/A	63–81 (19)
Touch Brush (K)	The sheep voluntarily contacts its nose/mouth to the brush.	N/A	82–90; 93–96 (13)
Brushed (L)	The sheep allows at least 3 front-to-back brushes with the brush across its head/back.	N/A	91–92; 97–99 (5)
	**Trial-Terminating (Undesired) Responses**		
Moving Away	Any movement of more than 0.25 m by a sheep away from the trainer.	N/A	N/A
Blocking Forrest	Any movement made by Fauna where ¾ or more of her body is between Forrest and the trainer. (Exp. 2 only)	N/A	N/A

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
