# Peer review of "Training Petting Zoo Sheep to Act Like Petting Zoo Sheep: An Empirical Evaluation of Response-Independent Schedules and Shaping with Negative Reinforcement"

_animals, 2020, doi:10.3390/ani10071122_

Round 1
Reviewer 1 Report
This manuscript presents the work of two training attempts to reduce sheep “fearful” behavior and increase approach in a petting zoo environment. The first experiment shows that an FT schedule of food was sufficient to create approach and reduce fear in one sheep. In the second experiment, negative reinforcement and positive reinforcement were combined and shown to effectively increase approach of the sheep in treatment, with little change for the control sheep.
Overall, this provides an interesting demonstration of applied behavior analysis in sheep. The sample size is small, but not unusual for this type of work. However, it does suggest that the work should be construed as more observational to the observed effects rather than experimental.
My primary concerns are that:
- The methods were hard to follow at several points. Perhaps because they were modified as what seemed necessary for behavior change, but I think more explanation for each step would help comprehension. More detailed questions are below.
- The FT success in Exp 1 seems to question the necessity of Neg reinforcement (or positive reinforcement) in Exp 2. It also seems as though FT is built into Exp 2 (throwing food at the start of the trial). To me, this makes it difficult to confidently ascribe which part of the treatment is the effective component.
Sentence structure lines 52-58 is hard to follow
Line 64, paradoxically seems awkward
Line 72, are you only referring to animal studies here?
Line 93- a Jacob sheep?
Line 94-97- this is confusing to me and hard to follow.
Methods: More information on the subjects would be useful. Their background, presenting issues, for how long, etc. Given that only 3 animals are used, a more case-study approach could be informative.
Based on Exp 1 results, its not clear why you wouldn’t repeat FT in other sheep (based on your explanation that perhaps Otis was just following the flock).
Why was FT a control in Exp2 if it seemed to work in Exp 1?
Exp 2, please clarify your trial terminating responses.
Clicker training-negative reinforcement- isn’t throwing food at the start the same as FT, which was shown to be effective in Exp 1? How do you know that is not the primary effective component of the package? Did Forrest have the same food thrown?
Figure 2- what are all the letters? What does j, K, l mean? The different conditions are hard to see. Can this be broken into two figures?
J-L are explained in later results. I think it would be useful to clarify your shaping step definitions in methods.
Line 347: what do you mean by “in reverse”
Line 364- I’m not sure if BAT is characterized as negative reinforcement typically
Line 412-414. I don’t think you quite tested neg reinforcement was the only way to change behavior.
Author Response
In addressing Reviewer 1's two main points:
1. We have created a General Methods section, as well as shortened and/or clarified some of the descriptions, which should help make the methods easier to read.
2. In Experiment 2, in 10 sessions (100 trials) of the FT/baseline condition, the target sheep, Fauna, never did more than face the trainer (step B) a handful of times, and move toward the trainer (step C) once. Therefore, it was unlikely that the FT schedule would have been effective at producing any desired change.
Also, the FT schedule was a baseline condition to test the effects of food itself. When the condition was changed to the clicker training conditions, food was only thrown at the start of a trial to test if the sheep would eat the food (during the FT conditions, both sheep in Experiment 2 rarely ate any food thrown at them). There otherwise was no FT schedule occurring during this condition. This is also why we provide data in the form of the "Eating" graph to show when the sheep did begin to eat the trial-starting food.
Specific edits are as follows:
Line 52-58 (now 50-57): edited.
"Paradoxically" removed.
Line 72 (shaping studies): No. Ghaemmaghami, Hanley, Jessel, & Landa, 2018, and Osborne & Himadi, 1990 are studies done with humans. I've added in one more reference that was just published (Fernandez & Dorey, 2020), but otherwise, this is every experimental examination (read: not just observational documentation) of shaping that has ever been done that we are aware of, including in the lab and with human or non-human subjects.
Line 93: Jacob sheep is the breed (they're often just called "Jacobs", like the way you might refer to the dog breed, "Australian shepherds", as "Aussies". Regardless, there's no need to mention this here, so we changed this to 'a sheep'.
Line 94-97 (final intro paragraph): Edited for readability.
Methods: I've included the following line about the subjects in the General Methods: "All three sheep were reported to avoid/escape the presence of all zoo staff and caretakers, as well as refuse to eat in the presence of any persons."
In terms of case-studies, I think what the reviewer might be thinking of is that this is not a between-subject study. This is what is described as a within- or single-subject experimental design, which is the common methodological approach to most behavior analytic and/or learning studies, like the subject matter of this special issue (i.e., Learning Theory Applied to the Welfare of Animals). Within-subject experimentation is not case-study, which is usually observation- and/or anecdote-based.
As noted above, the FT condition was not repeated as an intervention in Experiment 2 because it didn't work based on the 100 trials of baseline data presented. Throwing food at the start of a trial is not the same as an FT schedule because (a) it did not continue every 15 s, and (b) generally was not ate and was documented as such (see graph). The FT schedule, as well as the initial food thrown, was done as a control to show the effects (or lack thereof) for food.
Trial-terminating (undesired) responses are now described also as "undesired", and the response definitions along with the table have been moved to the General Methods.
Forrest did have the same trial-starting food thrown. I've added a line in the Experiment 2 Methods section to make that more clear (under each condition's description).
The letters in Figure 2 are described in Table 1. I've also added them now to the Figure 2 description. Also, now that there is a General Methods section, Table 1, with all shaping step responses, abbreviations, and definitions appear earlier in the manuscript and can be more easily referred to.
Line 347 (now 330): changed to "Similar procedures have been used to produce (rather than reduce)".
Line 364: BAT 2.0 avoids using the term negative reinforcement to describe itself. However, the original BAT book described itself as a negative reinforcement-based shaping procedure, citing this study (pre-manuscript, obviously) as the first type of study done on that same type of procedure.
Line 412-414: We removed the word "only".
Reviewer 2 Report
GENERAL COMMENTS
The manuscript provides novel data pertaining to shaping zoo sheep behaviour through differential reinforcement of successive approximations to a target response and the systematic measurement of such procedure. As the authors correctly indicate, this work has the potential to add useful information to the understanding of reinforcement training techniques which are often used in zoo animal management and husbandry practices.
This study may generate an interesting paper, and the manuscript has already very good structure, conciseness and consistency. I believe, however, it would benefit from some amendments to improve clarity and scientific soundness of the study findings.
MAJOR REMARKS
I believe there are some weaknesses to address in order to improve the manuscript:
- The Introduction would need more comprehensive and recent literature citations.
- The Introduction should include a final paragraph with hypotheses and predictions, which would also help to clarify the study aim(s).
- In Methods it is unclear whether potential confounding variables have been controlled for – particularly, potential effects of the isolation of the study sheep on results of Experiment 1; and changes of people involved over the data collection period in Experiment 2.
- The sample size (N=3 in total, over two experiments) is pretty small; I would advise to consider this major limitation in the Discussion of the experimental findings.
- Lack of statistical analyses; despite the small sample size, I believe it would be crucial to conduct some non-parametric statistical tests to evaluate the significance of the trends showed in the Results of both experiments.
- I would recommend separating Results and Discussion of specific experiments.
- I would also advise to cover the potential enriching effect of training techniques, such as Positive Reinforcement Training, in the Discussion
MINOR REMARKS
Please see below my comments:
- Abstract
- Lines 31-33: I would advise to reword this sentence to improve clarity
- Lines 34-36: Replace with “This study empirically examines…”
- Introduction
- Lines 51-54: Amend this to make it more consistent with the following points mentioned later in the sentence
- Line 53: It is Mandrillus sphinx, not Papio sphinx
- Line 64: Use “Surprisingly” (or similar term) rather than “Paradoxically”
- Line 80: Add “In addition,” (or similar phrase) before “One novel…”
- Line 90: Replace “we were effectively able to use” with “we used”
- Lines 94-97: Delete these and replace with clear study aim(s) as well as hypotheses & predictions
- Methods
- Line 111: Delete “several”
- Lines 179-180: Add “was”
- Results
- Line 188: Delete “Discussion”
- Lines 206-211: Move this down to Discussion section
- Lines 197, 199, 201, 202, 279, 281, 282, 293, 304, 309: Add figures (%)
- Line 218: Delete “another”
- Line 243: Delete “conducted”
- Line 262: Delete “Discussion”
- Lines 316-328: Move this down to Discussion section
- Discussion
- Line 336: Add figure (%)
- Line 364: Replace “create” with “established”
- Add considerations about potential enriching effects of PRT techniques (see papers such as Spiezio et al. 2015 Animal Welfare or Spiezio et al. 2017 Applied Animal Behaviour Science or their references)
Author Response
General edits:
Introduction: We've added a recent study published on an experimental examination of shaping (Fernandez and Dorey, 2020). This now brings the Introduction to 35 references, which includes every published study I am aware of that examines learning over time in shaping procedures with both human and non-human subjects, as well as many examples of non-within subject examinations (read: pre-/post-test effects) of shaping. 13/35 of these references are from the last 20 years.
The final paragraph of the Introduction has been edited to demonstrate what was tested. However, unlike hypothetico-deductive (between-subject) methodology, it is less common to list particular hypotheses for inductive, within-subject experimental methods.
A General Methods section has been added, which makes it more clear that the number of people involved in the study was the same for both experiments. The effects of otherwise testing the animals outside of their normal housing arrangement were accounted for by baseline conditions that demonstrated similar avoidance effects for the two sheep in experiment 2, and the fact that we could reverse this condition simply with food in experiment 1.
The sample size (N = 3) is standard for within-subject methodology. Within-subject methods within the field of behavior analysis (i.e., learning theory), particularly applied behavior analysis, do not typically incorporate inferential statistics (see the journals JEAB and JABA for both past and current examples). While I understand the reviewer's interest in applying some type of nonparametric inferential statistic to address these findings (I often do just that for other small N research), it is not necessary for the purposes of this manuscript, and potentially counterintuitive to a special issue dedicated to learning theory applied to the welfare of animals, which emphasizes in its special issue description, "Specific interest will be given to papers that use within-subject methodology to measure changes in behavior over time."
I have created a General Methods section to help clarify and separate the sections included in Experiments 1 and 2. For papers that involve multiple experiments, it is common to have major headings for each experiment (i.e., Experiment 1) that then combine a results and discussion section under that heading before arriving at a "General Discussion" section (see the following for an example: https://www.sciencedirect.com/science/article/pii/S0376635719303808).
I do like the idea of talking about the potential enrichment of PRT and do exactly that in other papers I have published (see, for instance, Fernandez and Timberlake, 2008; Fernandez, Kinley, and Timberlake, 2019). I do think, for the purposes of this paper, which is already a bit long, it may be less appropriate. I'm open to suggestions otherwise, but I think it would be better to leave the Discussion as is focused on (a) measuring shaping and other behavioral training procedures with animals, and (b) the ethics of using negative reinforcement with this and similar procedures and how that differs from "breaking" animals or the like.
Again, I'm certainly open to the idea, but for now, let's see what you think of the edits I've completed thus far. Please feel free to make specific suggestions if you still feel like this might benefit the paper after this revision.
Specific edits (based on previously submitted manuscript's lines):
Line 31-33: Changed to, "A changing-criterion design was used to empirically examine the effects of the shaping procedure during each step of the program."
Changed the last sentence of the Abstract to, "The result is one of the first studies to demonstrate the utility of using negative reinforcement within a shaping procedure to successfully intervene on approach/avoidance behaviors in an applied animal setting."
Lines 51-54: I'm not sure what edit you would like made here, but I've amended the entire second paragraph. Also, great catch on the correct genus and species; Papio was the old genus at the time Hal Markowitz published his paper in '78.
Line 64: Got rid of "Paradoxically" altogether. I'm happy to add "Surprisingly", but I think the sentence can stand on its own starting with, "Given".
Line 80: Changed to, "In addition, a novel approach..."
Line 90: Changed to, "we used".
Last paragraph of Introduction: I've adjusted the entire last paragraph accordingly. I do not include specific hypotheses/prediction, however, because that is less common for within-subject (inductive) experiments, as opposed to between-subject (hypothetico-deductive) methodology.
Line 111: Deleted "several".
Line 179-180: changed "were" to "was".
Results and Discussion: I've included a few more % numbers where applicable. I've changed the name of this section for both Experiment 1 and Experiment 2 to "Results and Discussion". However, as I noted above, it's common in multiple experiment papers to have a combined Results and Discussion section for each experiment, with a later General Discussion for the main points.
Line 218: Changed "another" to "a".
Line 243 (Exp. 2 baseline description): removed "conducted".
Line 364: Changed "created" to "established".
Reviewer 3 Report
Overall, this manuscript is extremely difficult to read and understand. The amount of superfluous language used dilutes the information trying to be conveyed and weakens the manuscript as a whole. I found the vague, meandering writing style to distract from the content, and found myself having to re-read many sections before understanding the meaning, with limited success. This could obviously be personal preference or a failing on my part, but this paper read more like a disjointed student submission than an academic publication.
In addition, I struggle to articulate the value of the current submission to the larger body of literature, possibly due in part to the writing quality. It is not clear what aspects of this project are novel or filling a gap in our knowledge. The introduction points out that there have been few systematic evaluations of a shaping procedure, as it is typically more “art than science”. The paper then goes on to illustrate this point as the planned shaping procedures and behavioral criteria were often adjusted based on the animal’s behavior. Taking out the data collection aspect, these case studies are unremarkable examples of successful application of behavior theory that animal trainers around the world employ every day. If anything, I would think that the addition of the trial protocols would have slowed progress toward desired behaviors, and made the trainer’s job more difficult. Though this is usually an acceptable trade-off in applied research, it is not clear what we’ve gained from this work. A better articulation of the purpose, goals, and potential applications of this research are needed before resubmission.
